# The Wnt/Beta-Catenin Pathway and Cytoskeletal Filaments Are Involved in the Positioning, Size, and Function of Lysosomes during Chick Myogenesis

**DOI:** 10.3390/cells11213402

**Published:** 2022-10-27

**Authors:** Kayo Moreira Bagri, Luiz Fernando Oliveira, Miria Gomes Pereira, José Garcia Abreu, Claudia Mermelstein

**Affiliations:** 1Laboratório de Diferenciação Muscular, Instituto de Ciências Biomédicas, Universidade Federal do Rio de Janeiro, Rio de Janeiro 21941-902, Brazil; 2Laboratório de Embriologia de Vertebrados, Instituto de Ciências Biomédicas, Universidade Federal do Rio de Janeiro, Rio de Janeiro 21941-902, Brazil; 3Laboratório de Ultraestrutura Celular Hertha Meyer, Instituto de Biofísica Carlos Chagas Filho, Universidade Federal do Rio de Janeiro, Rio de Janeiro 21941-902, Brazil

**Keywords:** lysosome, LAMP2, Wnt/beta-catenin, myogenesis, chicken

## Abstract

Lysosomes are highly dynamic organelles involved in the breakdown and recycling of macromolecules, cell cycle, cell differentiation, and cell death, among many other functions in eukaryotic cells. Recently, lysosomes have been identified as cellular hubs for the modulation of intracellular signaling pathways, such as the Wnt/beta-catenin pathway. Here we analyzed morphological and functional characteristics of lysosomes in muscle and non-muscle cells during chick myogenesis, as well as their modulation by the Wnt/beta-catenin pathway. Our results show that (i) muscle and non-muscle cells show differences in lysosomal size and its distribution, (ii) lysosomes are found in spherical structures in myoblasts and fibroblasts and tubular structures in myotubes, (iii) lysosomes are found close to the plasma membrane in fibroblasts and close to the nucleus in myoblasts and myotubes, (iv) lysosomal distribution and size are dependent on the integrity of microtubules and microfilaments in myogenic cells, (v) alterations in lysosomal function, in the expression of LAMP2, and in Wnt/beta-catenin pathway affect the distribution and size of lysosomes in myogenic cells, (vi) the effects of the knockdown of LAMP2 on myogenesis can be rescued by the activation of the Wnt/beta-catenin pathway, and (vii) the chloroquine Lys05 is a potent inhibitor of both the Wnt/beta-catenin pathway and lysosomal function. Our data highlight the involvement of the Wnt/beta-catenin pathway in the regulation of the positioning, size, and function of lysosomes during chick myogenesis.

## 1. Introduction

Skeletal muscle fiber formation is a highly complex process that occurs during embryonic development and postnatal life and involves several regulatory networks [1]. While the molecular and cellular mechanisms by which different signaling pathways regulate muscle development have been deeply analyzed in the last years, the interplay between specific cellular organelles, such as lysosomes, and signaling networks during myogenesis have been less studied.

Lysosomes are membrane-bound organelles that can vary considerably in size, shape, subcellular positioning, luminal pH, enzyme content, cargos, and function [2,3]. There are different lysosomal subpopulations in cells that vary considerably depending on the cell type [4]. Our understanding of the role of lysosomes in eukaryotic cells has considerably changed in the last years. Lysosomes are now seen as key components in the regulation of cell signaling, in addition to their well-established role in the degradation of molecules [5]. Both functions are highly intricate since the sequestration and degradation of signaling molecules can inhibit or activate specific signaling pathways, such as the Wnt/beta-catenin pathway. It has been reported that glycogen synthase kinase 3 (GSK3) can be sequestered from the cytosol into lysosomes, which activates the canonical Wnt/beta-catenin pathway [6]. In the non-activated state of the canonical Wnt/beta-catenin signaling pathway, GSK3 phosphorylates beta-catenin, triggering its degradation [7]. Thus, sequestration of GSK3 into lysosomes inhibits GSK3 activity leading to stabilization and activation of beta-catenin and TCF/LEF-dependent gene transcription, which activates the pathway.

Despite the recent increase in the understanding of the role of lysosomes in cell signaling, little is known about their role during skeletal muscle differentiation. Here, we studied morphological and functional features of lysosomes in muscle and non-muscle cells in chick myogenic cell cultures. Our results show the involvement of the canonical Wnt/beta-catenin pathway in the regulation of the positioning, size, and function of lysosomes in muscle cells.

## 2. Materials and Methods

### 2.1. Antibodies and Probes

Rabbit polyclonal antibody against the major lysosomal membrane protein LAMP2 (#PA1-655) was from ThermoFisher (São Paulo, Brazil). Mouse monoclonal antibodies against alpha-tubulin (clone DM1a) and rabbit polyclonal antibodies against beta-catenin (#C2206) were from Sigma-Aldrich (São Paulo, Brazil). Mouse monoclonal anti-Lamin A/C (#4777) was from Cell Signaling Technology (Danvers, MA, USA). NucSpot live cell nuclear stain was from Biotium Inc. (Fremont, CA, USA). DNA-binding probe 4,6-Diamino-2-phenylindole dihydrochloride (DAPI), Hoechst nuclear stain (#33342), Alexa Fluor 488-goat anti-rabbit IgG antibodies, and Alexa Fluor 546-goat anti-mouse IgG antibodies were from Molecular Probes (Eugene, OR, USA).

### 2.2. Cell Cultures

The primary cultures of myogenic cells were prepared from breast muscles of 11-day-old chicken embryos as previously described [8]. Chicken embryos were obtained from Granja Tolomei (Rio de Janeiro, Brazil) and handled according to the Institutional Animal Care and Use Committee, under protocol number 081/22. Briefly, fragments of pectoral muscle were incubated at 37 °C for 10 min in a calcium–magnesium-free solution (CMF, Sigma-Aldrich) containing 0.025% trypsin (Sigma-Aldrich). After removal of the trypsin solution, cells were dispersed by repeated pipetting in 8–1–0.5 culture medium (minimum essential medium with 10% horse serum, 0.5% chick embryo extract, 1% L-glutamine, and 1% penicillin/streptomycin, all from Invitrogen, Brazil). The resulting suspension was filtered, and cells were plated at an initial density of 7.5 × 10^5^ cells/35 mm culture dishes in 2 mL of medium. Cells were cultured on 22 mm-Aclar plastic coverslips (Pro-Plastics Inc., Linden, NJ, USA) previously coated with rat tail collagen. Cells were grown in a humidified 5% CO_2_ atmosphere at 37 °C. After the first 24 h, cultures were fed daily with fresh 8–1–0.5 cultured medium.

### 2.3. Cell Treatments

For lysosomal inhibition, 24 h chick myogenic cells were treated with the chloroquine Lys05 (Sigma-Aldrich, dissolved in 0.1% DMSO) at a final concentration of 1 μM; or with 0.1% DMSO, both for 24 h. For the activation of the Wnt/beta-catenin pathway, 24-h cells were treated with 5 µM 6-bromoindirubin-30-oxime (BIO, Sigma-Aldrich) or conditioned media (50% *v*/*v*) enriched in Wnt3a (obtained from L-Wnt3a cells, ATCC, Manassas, VA, USA). For acidic cellular organelle labeling, cells were stained with 2 ng/mL acridine orange solution for 10 min at 37 °C and 5% CO_2_. The cytoskeleton interfering drugs Taxol (1 µM), Colcemid (0.05 µM), and Cytochalasin B (0.1 µM), all from Sigma-Aldrich, were used in 24 h cells.

### 2.4. LAMP2 Knockdown

siRNA sequences specific for LAMP2 of *Gallus gallus* (mRNA sequence obtained from NCBI reference sequence NM_001030295) or enhanced green fluorescent protein (eGFP) were from Integrated DNA Technologies (Coralville, IA, USA). The *Gallus gallus* sequences of siRNA that were selected to target LAMP2 or eGFP are:

eGFP from *Gallus gallus;*5′ rArArGrCrUrGrArCrCrCrUrGrArArGrUrUrCrArUrCrUrGCA 3′ (strand);5′ rUrGrCrArGrArUrGrArArCrUrUrCrArGrGrGrUrCrArGrCrUrUrGrC 3′ (passenger);LAMP2 from *Gallus gallus* (seq 1467-1492);sense 5′ rGrArArUrUrGrGrArArArUrUrUrArCrUrUrGrArArGrCrUAC 3′;antisense 5′ rGrUrArGrCrUrUrCrArArGrUrArArArUrUrUrCrCrArArUrUrCrArU 3′.

### 2.5. Wnt/Beta-Catenin Gene Reporter Assay

RKO-pBAR/Renilla (RKO B/R) or SW480-pBAR/Renilla (SW480 B/R) cells were seeded into 96-well plates with 2.0 × 10^4^ cells per well. Cells were treated with increased concentrations of Lys05 (0.01, 0.1, 1, and 10 µM) for 24 h. After treatment, cells were lysed with passive lysis buffer (Promega, Madison, WI, USA) and assayed for Firefly and Renilla luciferase activity using the dual-luciferase reporter assay (Promega #E1960). Luminescence was measured using the Modulus IITM Microplate Multimode Reader (Promega). Control media (DMEM F-12) were used as negative controls for Wnt/beta-catenin reporter stimulation by BIO or CHIR99021 in RKO B/R. All experiments were performed in triplicate and repeated at least three times.

### 2.6. Immunofluorescence and Digital Image Acquisition

Chick myogenic cells were rinsed with PBS and fixed in absolute methanol at −20 °C for 5 min. Then, they were permeabilized and blocked in PBS with 0.1% saponin, 1% BSA, and 5% horse serum for 30 min. Cells were incubated overnight at 4 °C with primary antibodies (all diluted 1:50 in the permeabilization and blocking solution). After incubation, cells were washed for 30 min in the permeabilization and blocking solutions and incubated for 1 h at 37 °C with Alexa Fluor-conjugated secondary antibodies (all diluted 1:100 in the permeabilization and blocking solution). Nuclei were labeled with DAPI, Hoechst, or NucSpot. Cells were mounted in Prolong Gold (Molecular Probes) and examined with either an Axiovert 100 microscope (Carl Zeiss, Jena, Germany) coupled to an Olympus DP71 high-resolution camera, a Leica TCS SPE laser scanning confocal microscope (Leica, Wetzlar, Germany), or a DSU Spinning Disk confocal scanner mounted on an inverted fluorescent microscope (Olympus, Shinjuku, Japan). Control experiments with no primary antibodies showed only a faint background staining (data not shown). Image processing was performed using Fiji software, version 1.53q [9] and figure plates were mounted with Adobe Photoshop software, version 7.0.1 (Adobe Systems Inc., Mountain View, CA, USA).

### 2.7. Quantification of Chick Cell Cultures

The nuclear localization of beta-catenin, the diameter of myotubes, the diameter and area of lysosomes, as well as the distance between lysosomes and the nuclei, were quantified using Fiji software, https://imagej.net/software/fiji/ (accessed on 25 August 2022). We measured the total area occupied by lysosomes in the three phenotypes of cultured cells (myoblasts, myotubes, and fibroblasts) by analyzing images of Lysotracker (lysosomal marker), Hoechst (nuclear marker), and phase contrast microscopy (whole cell image). The limits of cells were accessed by phase contrast microscopy. The area occupied by lysosomes per cell was measured concerning the total cell area (area of lysosomes (%) ÷ total cell area (%)). DAPI and desmin double staining enables the identification of myoblasts through the presence of desmin and fibroblasts through differences in nuclear morphology and fluorescence intensity [10]. More specifically, muscle fibroblasts show large, flattened, and pale nuclei and negative cytoplasm staining for desmin, whereas myoblasts are desmin-positive cells with small, round, and bright nuclei [10]. All data were quantified from at least fifty randomly chosen microscopic fields collected from three independent experiments.

### 2.8. Nuclear and Cytoplasmic Fractions

Nuclear and cytoplasmic fractions were prepared based on a previously described method [11]. Cultured chicken myogenic cells were scraped off the dishes with 500 μL of culture medium, transferred to Eppendorf tubes, and centrifuged at 4000 rpm for 10 min. The supernatant was discarded, and cells were washed in sterile PBS, then centrifuged at 4000 rpm for 10 min. Then, cells were resuspended in 500 μL of hypotonic buffer solution (20 mM Tris-HCl pH 7.4, 10 mM KCl, 2 mM MgCl_2_, 1 mM EGTA, 0.5 mM DTT, and 0.05% sodium azide) containing 0.1% NP-40 and protease inhibitor cocktail (#P2714, Sigma-Aldrich), vortexed for 5 s and incubated for 3 min on ice cold. Then, cells were transferred to a Dounce-type homogenizer and macerated 20 times. Subsequently, the cell extract was centrifuged at 13,400 rpm for 10 min, fractionating the nucleus (pellet) and cytoplasm (supernatant). The pellet was resuspended in hypotonic buffer solution, and then the pellet and the supernatant were centrifuged again at 13,400 rpm for 10 min, and the cycle was repeated three times. For immunoblot analysis, 100 μL of RIPA buffer (150 mM sodium chloride, 50 mM Tris-HCl pH 8.0, 1% NP-40, 0.5% sodium deoxycholate, 0.1% SDS), protease inhibitor cocktail, and DNAse were added to each sample.

### 2.9. Immunoblotting

The protein concentration in cytoplasmic and nuclear fractions was determined with the Bradford assay and denatured with a Laemmli buffer at 95 °C for 5 min. Samples were separated on 10% sodium dodecyl sulfate–polyacrylamide gel electrophoresis (SDS-PAGE) and transferred onto 0.45 μm polyvinylidene fluoride (PVDF) membranes (Millipore, Burlington, MA, USA, #IEVH85R). After the blockade of nonspecific protein signals in 2% (*w*/*v*) polyvinylpyrrolidone (PVP) (Sigma-Aldrich, #P0930) in Tris-buffered saline with 0.01% Tween-20 (TBST) for 1 h, PVDF membranes were incubated overnight at 4 °C with primary antibodies against α-tubulin (1:2000, Sigma-Aldrich, #T9026), beta-catenin (1:2000, BD Bioscience, #610154), Lamin A/C (1:500, Cell Signaling Technology, #4777) and desmin (1:2000, Sigma-Aldrich, #D8281). The next day, membranes were incubated with horseradish peroxidase-conjugated secondary antibodies (1:5000, Invitrogen, #31460 and #31430) and antibody-bound bands were visualized with the SuperSignal™ West Pico PLUS Chemiluminescent Substrate using a ChemiDoc™ XRS+ System with Image Lab™ Software, version 6.1 (Bio-Rad Laboratories, Hercules, CA, USA #1708265).

### 2.10. Statistical Analysis

Statistical analysis was carried out using GraphPad Prism software version 8. The results of at least three independent experiments were compared. Statistical analysis was performed with a one-way ANOVA followed by a Tukey’s multiple comparison test, * *p* < 0.05, ** *p* < 0.01, *** *p* < 0.001, and **** *p* < 0.0001. Error bars denote mean ± SEM.

## 3. Results

### 3.1. Muscle and Non-Muscle Cells Show Differences Regarding Lysosomal Size and Distribution

To understand the role of lysosomes during skeletal myogenesis, we analyzed several morphological and functional characteristics of lysosomes in primary cultures of chick myogenic cells. First, we examined the intracellular distribution, and the distance to the nucleus, size, and area of lysosomes in the three main cellular phenotypes that are characteristic of chick myogenic cultures, which are mononucleated myoblasts, multinucleated myotubes, and fibroblasts [10]. Chick myogenic cells were grown for 24, 48, 72, and 96 h and labeled with lysotracker and the nuclear dye Hoechst (Figure 1A–D). The quantification of the diameter and area of lysosomes along myogenesis and between the three different cell phenotypes showed that there was an increase in 48 h cultures in the diameter of lysosomes in myoblasts and myotubes, compared with other time points of muscle cell cultures (Figure 1F). The percentage of the total area occupied by lysosomes showed an increase over time (from 24 to 72 h) in the three cell phenotypes. Curiously, myotubes showed higher values of the lysosomal area in all time points, in comparison with myoblasts and fibroblasts, reaching as much as ~60% of the area of cells in 96 h (Figure 1E). The quantification of the distance of lysosomes to the nucleus in each cell phenotype showed that lysosomes from fibroblasts were 2- to 6-times more distant from the cell nucleus than those from myoblasts and myotubes in all time points of the cell cultures (Figure 1G). Indeed, lysosomes from myotubes and myoblasts were found in the perinuclear region of cells, whereas lysosomes from fibroblasts were located near the plasma membrane (Figure 1A–D). We also analyzed the intracellular distribution of lysosomes in myogenic cells by using an anti-LAMP2 antibody. Immunofluorescence labeling of LAMP2 in 48 h chick muscle cells showed a distinct distribution of lysosomes in myoblasts, fibroblasts, and myotubes (Figure 2). Although most of the lysosomes were spherical in myoblasts and fibroblasts, they were found as long tubular structures in multinucleated myotubes (Figure 2). Furthermore, myotubes have more lysosomes as compared to myoblasts and fibroblasts.

### 3.2. Lysosomal Size and Distribution Are Dependent on the Integrity of Microtubules and Microfilaments in Myogenic Cells

Since we found tubular lysosomal structures in myotubes and these structures were described as dependent on a network of microtubules in cells [12], we decided to test whether cytoskeletal filaments were involved in lysosomal structures and dynamics in chick muscle cells. Chick myogenic cells were grown for 24 h, treated with agents that interfere with cytoskeletal polymerization and/or stability (Cytochalasin B, Colcemid, or Taxol), and labeled with acridine orange and DAPI. Our results show that the disorganization of microtubules (Colcemid and Taxol) or microfilaments (Cytochalasin B) led to an increase in lysosomal compartments (Figure 3), suggesting that microtubules and microfilaments are involved in lysosomal dynamics in chick muscle cells.

### 3.3. Alterations in Lysosomal Function and Wnt/Beta-Catenin Pathway Change the Distribution and Size of Lysosomes in Myogenic Cells

Next, we decided to analyze whether alterations in lysosomal function could lead to changes in the distribution and size of lysosomes in myogenic cells. To interfere with lysosomal function, we treated chick myogenic cells with the lysosomal inhibitor Lys05 and found that the chloroquine Lys05 reduced the distance of lysosomes to the nucleus and increased the size and area of lysosomes in myoblasts, fibroblasts, and myotubes (Figure 4). We also analyzed the impact of the Wnt/beta-catenin pathway on the distribution and size of lysosomes in myogenic cells. Treatment of cells with activators of the Wnt/beta-catenin pathway (BIO and Wnt3a) induced proximity of lysosomes to the nuclei and increased the size and area of lysosomes in myoblasts, fibroblasts, and myotubes (Figure 4). Interestingly, the two activators of the Wnt/beta-catenin pathway (BIO and Wnt3a) were able to revert the effects of Lys05 on the distribution and size of lysosomes in chick muscle cells (Figure 4).

### 3.4. Lys05 Inhibits the Wnt/Beta-Catenin Transcriptional Reporter

To elucidate Lys05 effects on Wnt/beta-catenin signaling activity, we stimulated the RKO-pBAR/Renilla reporter cells with 2 µM BIO or 3 µM CHIR99021 and co-treated these cells with 0.01, 0.1, 1, and 10 µM of Lys05 for 24 h. We observed that 10 µM Lys05 significantly inhibited Wnt/beta-catenin pathway transcriptional reporter activity in cells stimulated with both BIO (Figure 5) and CHIR99021 (Figure 5B), two small molecules that activate the Wnt/beta-catenin signaling pathway through inhibition of glycogen synthase kinase (GSK3). To strengthen this observation, we employed SW480-pBAR/Renilla cells, which harbor an APC mutation that disassembles the destruction complex, resulting in constitutive activation of beta-catenin. Interestingly, Lys05 also inhibits the Wnt/beta-catenin transcriptional reporter activity in a concentration-dependent manner in these cells (Figure 5C). The gene reporter assay data suggest that Lys05 inhibits the Wnt/beta-catenin signaling pathway, especially with 10 µM (Figure 5D).

### 3.5. Knockdown of LAMP2 Can Be Rescued by the Activation of the Wnt/Beta-Catenin Pathway

Since we found a correlation between the Wnt/beta-catenin pathway and lysosomal positioning and size, we decided to further explore this relationship by analyzing the intracellular distribution of beta-catenin in muscle cells (Figure 6A–G). Chick myogenic cells were grown for 24 h and transfected with siRNA against LAMP2. Our group previously showed the efficacy of the knockdown of siRNA sequences specific for LAMP2 of *Gallus gallus* in a previous paper from our lab [13]. In this previous work, we showed via Western blot that the transfection of chick muscle cells with siRNA against LAMP2 induced ~70% reduction in LAMP2 protein levels. Beta-catenin localization was assessed by immunofluorescence microscopy of chick muscle cells treated with siRNA against LAMP2, eGFP, Wnt3a, and BIO. Beta-catenin was found in the cytoplasm of cells transfected with siRNA against LAMP2 and eGFP, and in the control (untreated cells), whereas treatment with Wnt3a e BIO induced its nuclear translocation (Figure 6A–G). Interestingly, the knockdown of LAMP2 induced an inhibition in the formation of myotubes, but Wnt3a and BIO were able to block these effects (Figure 6H). Wnt3a and BIO were also able to induce the nuclear translocation of beta-catenin in cells transfected with siRNA against LAMP2 (Figure 6I). Western blot analysis of nuclear and cytoplasmic fractions isolated from chick myogenic cells was used to further confirm the intracellular localization of beta-catenin (Figure 7). Beta-catenin was found predominantly in the cytoplasmic fraction of muscle cells and with a minor amount detected in the nuclear fraction (Figure 7), which is in accordance with the results shown in Figure 7. In the nucleus/cytoplasm experiments, Lamin A/C was used as a nuclear protein marker and desmin as a cytoplasmic marker (Figure 7). Interestingly, desmin was found in both the cytoplasmic and nuclear fractions (with a greater amount in the cytoplasm). Desmin is a muscle-specific intermediate filament protein, and its filaments are, in addition being found in the cytoplasm of muscle cells, stably associated with the outer nuclear surface of chick muscle cells [14], which explains the presence of desmin in both cytoplasmic and nuclear fractions.

## 4. Discussion

The lysosomal system has been implicated in several aspects of muscle structure and physiology, such as the control of myofiber homeostasis and muscle mass [15]. However, the molecular and cellular mechanisms involved in the interplay between lysosomes and signaling pathways during myogenesis are not known. Here, we aimed to explore different mechanisms that could modulate the positioning, size, and function of lysosomes during myogenesis, including the Wnt/beta-catenin pathway. We used a primary muscle cell culture system to study the morphological and functional features of lysosomes. These cell cultures were obtained from the pectoral skeletal muscle of 11-day-old chick embryos (*Gallus gallus*), and they harbor three main cell phenotypes: myoblasts, myotubes, and fibroblasts [10,13]. The chick myogenic cell culture is a robust model of myogenesis, in which myotubes can harbor hundreds of nuclei.

The collection of our results shows that (i) lysosomal size distribution and its distribution size are different when comparing muscle and versus non-muscle cells, (ii) lysosomes are found in spherical structures in myoblasts and fibroblasts and tubular structures in myotubes, (iii) lysosomes are found close to the plasma membrane in fibroblasts and close to the nucleus in myoblasts and myotubes, (iv) lysosomal distribution and size are dependent on the integrity of microtubules and microfilaments in myogenic cells, (v) alterations in lysosomal function, expression of LAMP2, and Wnt/beta-catenin signaling pathway affects the distribution and size of lysosomes in myogenic cells, (vi) the effects of the knockdown of LAMP2 on myogenesis can be rescued by the activation of the Wnt/beta-catenin pathway, and (vii) the chloroquine Lys05 is a potent inhibitor of both the Wnt/beta-catenin pathway and lysosomal function.

Lysosomes are generally described as small spherical organelles, but they may form several microns-long tubules that interconnect as a network [12,16,17]. It has been shown that spherical lysosomes can spontaneously elongate into tubules in certain conditions [18]. Chemical imaging revealed that tubular lysosomes differ from vesicular ones in terms of their luminal pH, calcium, and proteolytic activity [18]. Here, we found lysosomes as spherical structures in myoblasts and fibroblasts and as tubular structures in myotubes. We can hypothesize that during muscle differentiation, the vesicular lysosomes found in myoblasts elongate and fuse into tubules in myotubes and that this morphological change is accompanied by a change in their function.

Lysosomal size is tightly controlled in cells and is intimately correlated with lysosomal function [2]. Lysosomal diseases are frequently accompanied by alterations in lysosomal size. The enlarged lysosomes that we detected in chick muscle cells after the inhibition of lysosomal function with Lys05 are probably related to acidification defects that promote the accumulation of undegradable cargo in catabolically inactive lysosomes, as previously described [2]. Interestingly, we found an increase in the diameter of lysosomes in myoblasts from 48 h-cultures, which could be related to the peak of myoblast fusion in chick muscle cultures. These results agree with previous data indicating that most chick myoblast fusion occurs between 48 and 65 h in vitro [10,19].

Lysosomes were found close to the plasma membrane in fibroblasts, whereas they were close to the nucleus in myoblasts and myotubes. It has been reported that the intracellular positioning of lysosomes correlates with their luminal pH and function [20]. Since peripheral lysosomes (near the plasma membrane) have been associated with reduced acidification and impaired proteolytic activity [20], we can infer that chick myoblasts and myotubes have lysosomes with increased proteolytic activity, as compared to muscle fibroblasts. Furthermore, lysosomes have been implicated in the regulation of different signaling pathways, such as mTOR and Wnt/beta-catenin [3]. Regarding the relationship between lysosomes and the mTOR pathway, in the presence of amino acids, mTORC1 is located on peripheral lysosomes close to its upstream signaling elements, whereas during starvation mTORC1 and lysosomes are preferentially clustered in the perinuclear area, facilitating fusion of lysosomes with autophagosomes [3].

The intracellular positioning of lysosomes is dependent on their movement within the cell which has been associated with microtubules [12,21]. Here, we showed that the distribution of lysosomes in myogenic cells was altered in the presence of agents that interfere with the polymerization or stability of microtubules and microfilaments. Interestingly, these cytoskeletal interfering drugs induced an increase in the size of lysosomes suggesting that the cytoskeleton regulates the movement, size, and function of lysosomes in muscle cells. Importantly, it has been reported that nocodazole, but not Colchicine or Taxol, alters lysosome acidity and induces lysosomal disruption [22]. Therefore, the effects of Taxol and Colcemid on lysosomes from muscle cells that we described here need to be further explored.

LAMP1 and LAMP2 are major lysosomal membrane proteins involved in lysosomal biogenesis and in the maintenance of the structural integrity of the lysosomal compartment [23]. Our data show that the knockdown of LAMP2 by siRNA inhibited the formation of myotubes and altered the distribution and size of lysosomes in chick muscle cell cultures. These results agree with previous data showing that LAMP1 or LAMP2 depletion by siRNA impaired the differentiation of C2C12 mouse myoblasts, reduced the diameter of C2C12 myotubes, and decreased the expression levels of myogenic regulatory factors, MyoD, and myogenin [24].

Importantly, our results show the involvement of the Wnt/beta-catenin pathway in the regulation of the lysosomal morphology and function during chick myogenesis. Two activators of the Wnt/beta-catenin pathway, Wnt3a and BIO, were able to rescue the deleterious effects of the knockdown of LAMP2 in the formation of muscle fibers. Different mechanisms could be involved in the interplay between lysosomes and the Wnt pathway. It has been reported that GSK3 can be sequestered from the cytosol into lysosomes, which activates the canonical Wnt/beta-catenin pathway [6]. Further studies are necessary to understand the molecular mechanism involved in the Wnt/beta-catenin regulation of lysosomal dynamics in muscle cells.

## 5. Conclusions

In conclusion, our data unveil new morphological and functional features of lysosomes during chick myogenesis and provide new insights into the role of lysosomes in the modulation of the Wnt/beta-catenin pathway in muscle.

## Figures and Tables

**Figure 1 cells-11-03402-f001:**
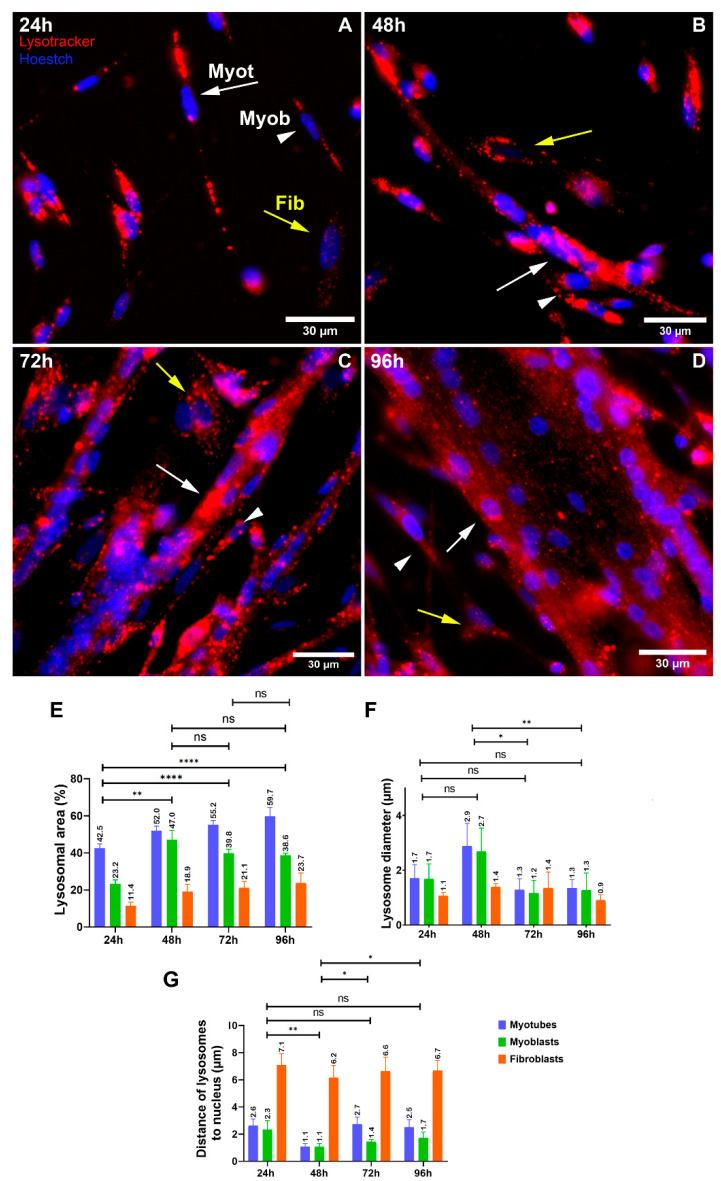
Lysosomal distribution and size differ between muscle and non-muscle cells. Chick myogenic cells were grown for 24, 48, 72, and 96 h and labeled with Lysotracker and Hoechst (**A**–**D**). White arrows represent myotubes (Myot), white arrowheads represent myoblasts (Myob), and yellow arrows represent fibroblasts (Fib). Scale bars in (**A**–**D**) = 30 µm. The area and diameter of lysosomes, and the distance of lysosomes to the nucleus, were quantified from at least fifty randomly chosen microscopic fields collected from three independent experiments. One-way ANOVA followed by Tukey’s multiple comparison test, * *p* < 0.05, ** *p* < 0.01, **** *p* < 0.0001. Error bars denote mean ± SEM. n.s. = non-significant.

**Figure 2 cells-11-03402-f002:**
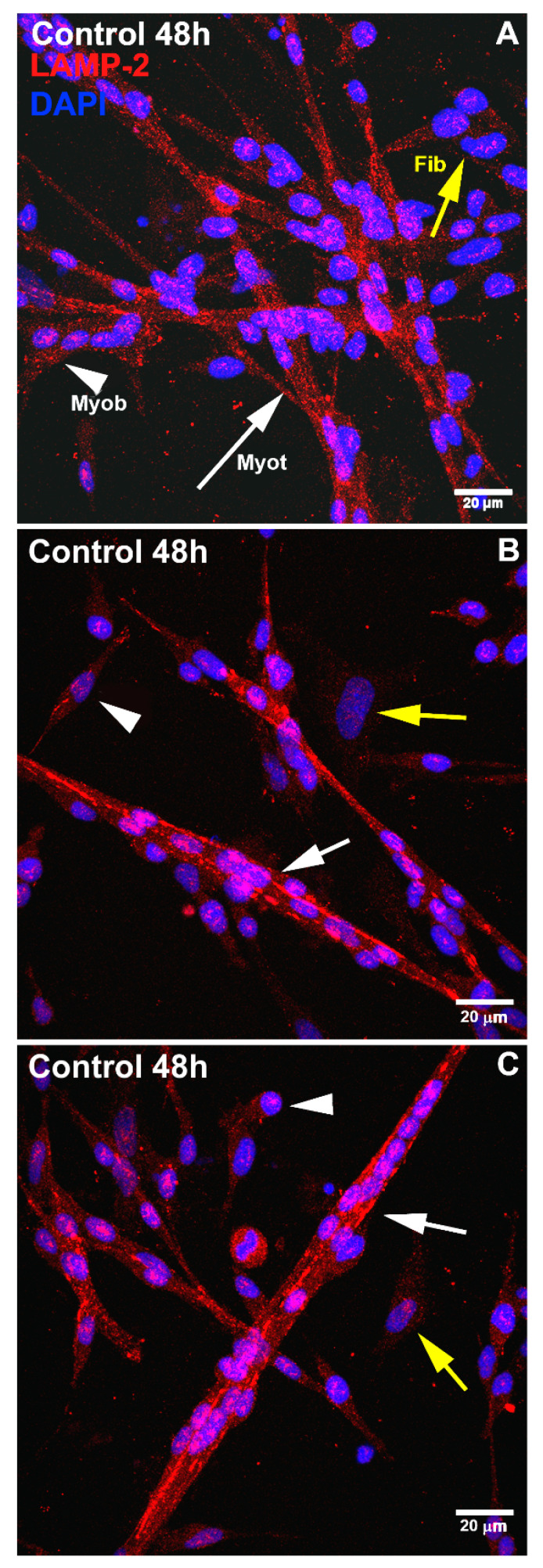
LAMP2 distribution changes during chick myogenesis. Chick muscle cells were grown for 48 h and double-labeled with antibodies against LAMP2 (red, **A**–**C**) and with the nuclear dye DAPI (blue, **A**–**C**). The three images represent myogenic cultures grown for 48 h (**A**–**C**). White arrows represent myotubes (Myot), white arrowheads represent myoblasts (Myob), and yellow arrows represent fibroblasts (Fib). LAMP2 labeling was found in a tubular distribution in the cytoplasm of multinucleated myotubes (Figure 2). Furthermore, myotubes showed increased labeling of LAMP2 as compared to myoblasts and fibroblasts. LAMP2 labeling in fibroblast cells was weak. Scale bars in (**A**–**C**) = 20 µm.

**Figure 3 cells-11-03402-f003:**
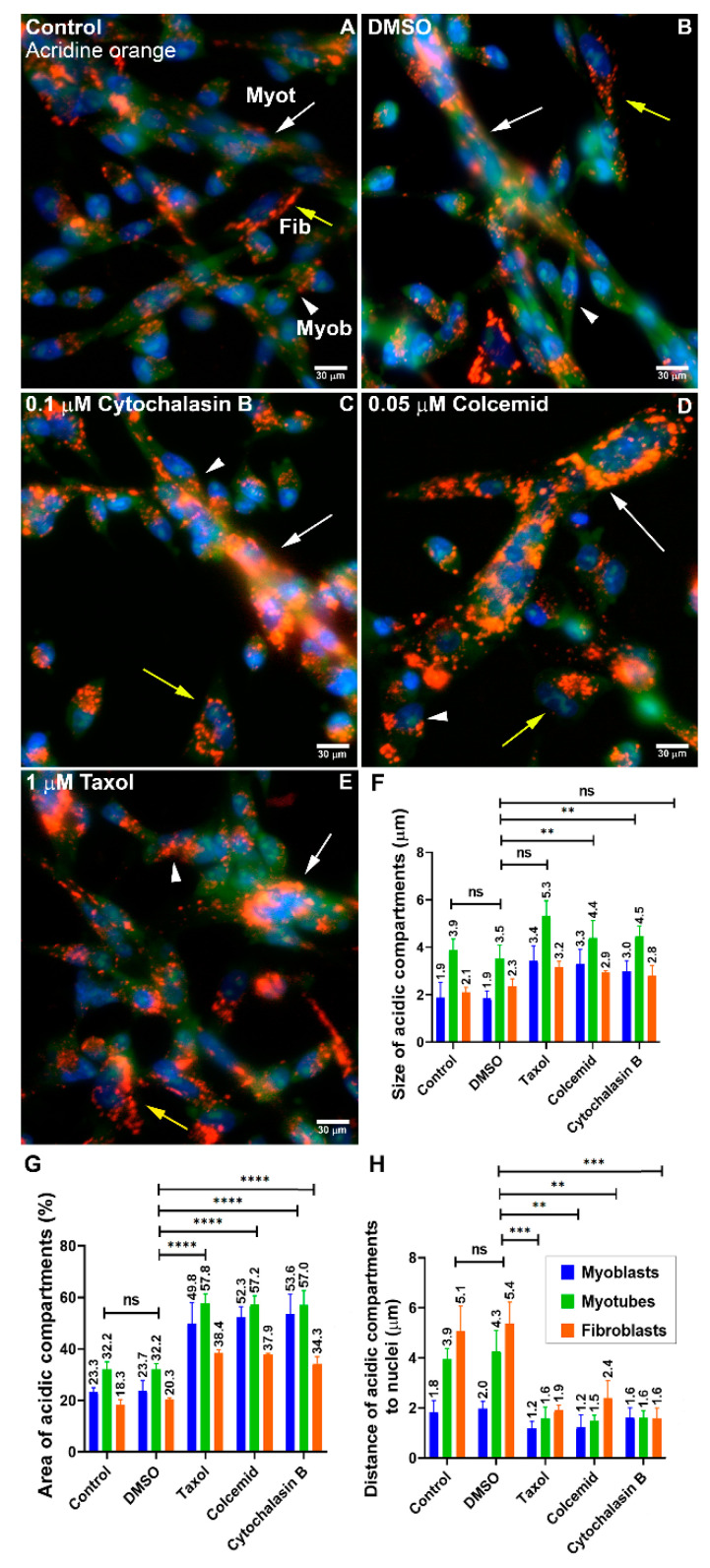
Effects of the disorganization of microtubules and microfilaments during chick myogenesis. Chick muscle cells were grown for 24 h and treated with Cytochalasin B, Colcemid, and Taxol for the next 24 h. Cells were labeled with acridine orange (**A**–**E**). White arrows represent myotubes (Myot), white arrowheads represent myoblasts (Myob), and yellow arrows represent fibroblasts (Fib). Note the increase in acridine orange labeling in cells treated with Cytochalasin B (0.1 µM), Colcemid (0.05 µM), and Taxol (1 µM). Scale bars in (**A**–**E**) = 30 µm. The distance to nuclei, size, and area of acridine orange-positive structures (acidic compartments) was quantified from at least fifty randomly chosen microscopic fields collected from three independent experiments (**F**–**H**). One-way ANOVA followed by Tukey’s multiple comparison test, ** *p* < 0.01, *** *p* < 0.001, **** *p* < 0.0001. Error bars denote mean ± SEM. n.s. = non-significant.

**Figure 4 cells-11-03402-f004:**
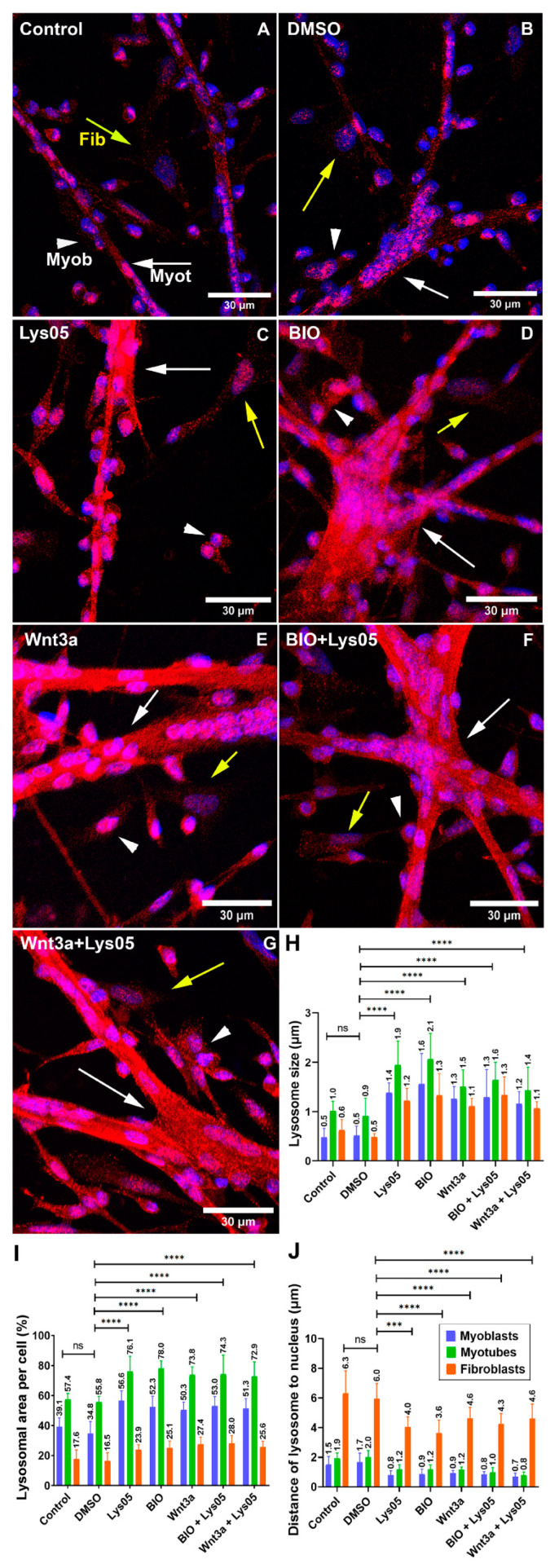
Activation of the Wnt/beta-catenin pathway increases the lysosomal compartment in chick muscle cells. Chick myogenic cells were grown for 24 h and treated with Lys05, BIO, Wnt3a, BIO + Lys05, or Wnt3a + Lys05. Cells were double-labeled with antibodies against LAMP2 and with the nuclear dye DAPI (**A**–**G**). White arrows represent myotubes (Myot), white arrowheads represent myoblasts (Myob), and yellow arrows represent fibroblasts (Fib). Scale bars in (**A**–**G**) = 30 µm. The following parameters were analyzed: the size of lysosomes, the total area occupied by lysosomes in cells, and the distance of lysosomes to the nucleus (**H**–**J**). Quantifications were performed from at least fifty randomly chosen microscopic fields collected from three independent experiments. One-way ANOVA followed by Tukey’s multiple comparison test, *** *p* < 0.001, **** *p* < 0.0001. Error bars denote mean ± SEM. n.s. = non-significant.

**Figure 5 cells-11-03402-f005:**
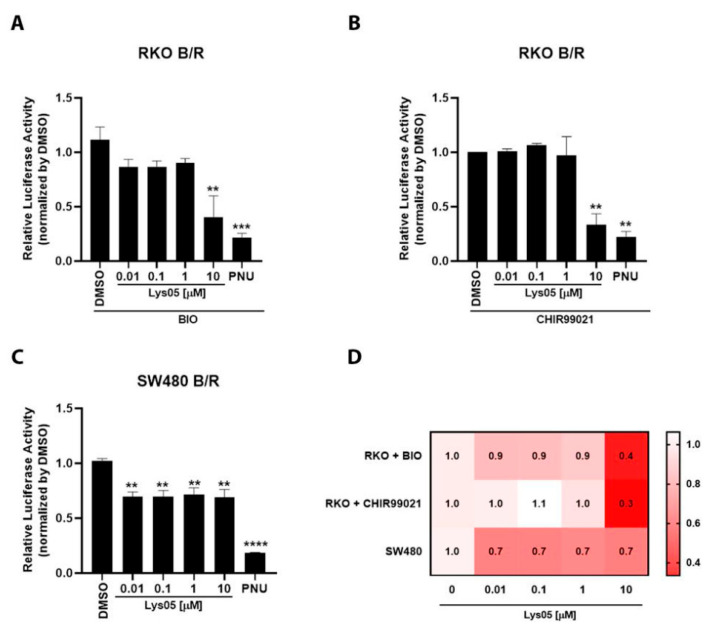
Lys05 inhibits the Wnt/beta-catenin transcriptional reporter. Cells were treated with the DMSO as vehicle control, Lys05 (0.01, 0.1, 1, and 10 µM), and 100 µM PNU as the negative control for 24 h. Lys05 significantly inhibited transcriptional reporter activity of the Wnt/beta-catenin signaling pathway in RKO B/R cells stimulated with both 2 µM BIO (**A**) and 3 µM CHIR99021 (**B**). Lys05 also inhibited the Wnt/beta-catenin pathway transcriptional reporter in SW480 B/R cells (**C**). The heatmap summarizes the effects of Lys05 by concentration and cell type (**D**). *n* = 3, performed in triplicate. One-way ANOVA followed by Tukey’s multiple comparison test, ** *p* < 0.01, *** *p* < 0.001, **** *p* < 0.0001). Error bars denote mean ± SEM.

**Figure 6 cells-11-03402-f006:**
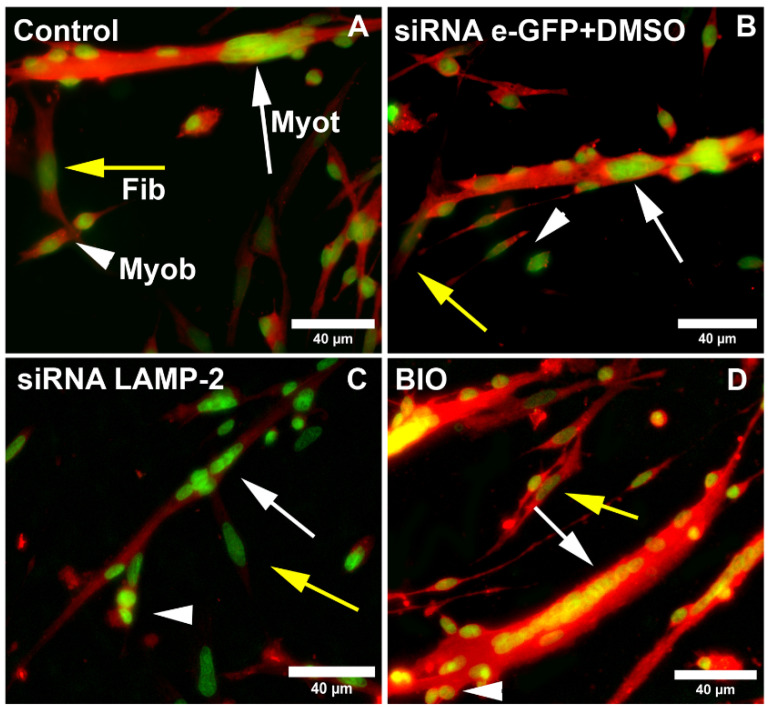
Activation of the Wnt/beta-catenin pathway can revert the effects of the downregulation of LAMP2 in chick muscle cells. Chick myogenic cells were grown for 24 h and either transfected with siRNA against LAMP2 or treated with BIO or Wnt3a. Cells were double-labeled with antibodies against beta-catenin (red, **A**–**G**) and the nuclear dye NucSpot (green, **A**–**G**). White arrows represent myotubes (Myot), white arrowheads represent myoblasts (Myob), and yellow arrows represent fibroblasts (Fib). Myotube diameter decreased after the downregulation of LAMP2, whereas both Wnt3a and BIO were able to revert this effect (**H**). Beta-catenin labeling was found in the cytoplasm in untreated (control) cells and in cultures transfected with siRNA against eGFP and against LAMP2 (**A**–**G**). Note the decrease in the nuclear localization of beta-catenin in myotubes after the downregulation of LAMP2, which was reverted by BIO and Wnt3a (**I**). Scale bars in (**A**–**G**) = 40 µm. Quantifications were performed from at least fifty randomly chosen microscopic fields collected from three independent experiments. One-way ANOVA followed by Tukey’s multiple comparison test, * *p* < 0.05, ** *p* < 0.01, *** *p* < 0.001, **** *p* < 0.0001. Error bars denote mean ± SEM. n.s. = non-significant.

**Figure 7 cells-11-03402-f007:**
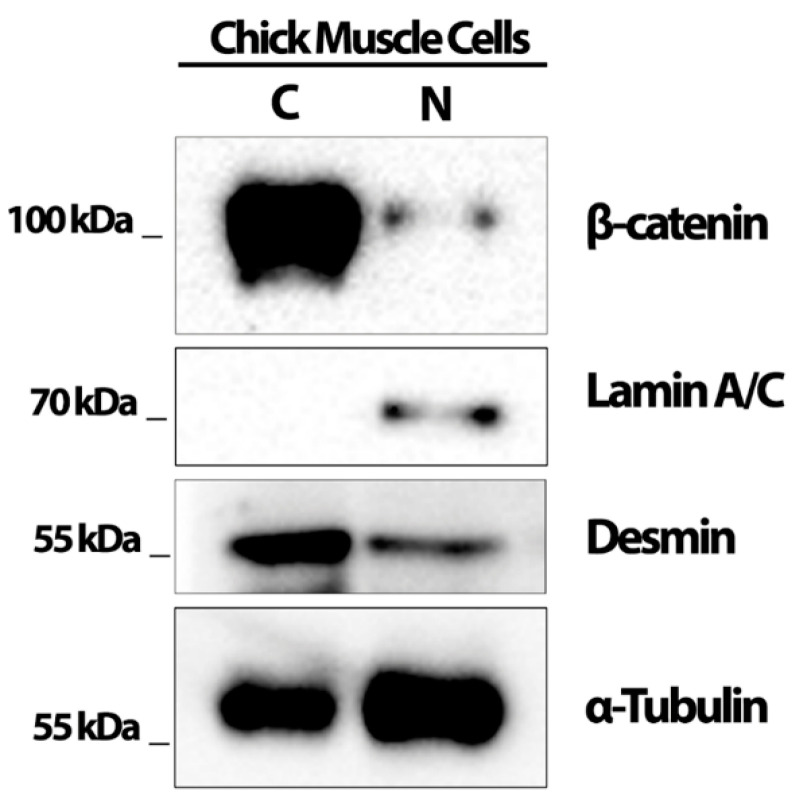
Beta-catenin is predominantly found in the cytoplasm of chick muscle cells. Cytoplasmic (C) and nuclear (N) fractions were isolated from chick myogenic cells grown for 48 h. Nuclear and cytoplasmic fractions were submitted to SDS-PAGE and immunoblot against beta-catenin, lamin A/C, desmin, and alpha-tubulin. Note the presence of lamin A/C exclusively in the nuclear fraction, whereas beta-catenin was predominantly detected in the cytoplasmic fraction.

## Data Availability

The datasets generated for this study can be found within the manuscript figures.

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
