# Peer review of "The Wnt/Beta-Catenin Pathway and Cytoskeletal Filaments Are Involved in the Positioning, Size, and Function of Lysosomes during Chick Myogenesis"

_cells, 2022, doi:10.3390/cells11213402_

Round 1
Reviewer 1 Report
The topic is highly relevant to the scientific field and to the technological advance for the development of new therapies for human and animal health. The article is well-written, clear, and the experimental design is adequate. Therefore, I recommend accepting the manuscript for publication.
Author Response
Reviewer: 1
1 - The topic is highly relevant to the scientific field and to the technological advance for the development of new therapies for human and animal health. The article is well-written, clear, and the experimental design is adequate. Therefore, I recommend accepting the manuscript for publication.
Author’s response: We thank the reviewer for the analysis and comments.
Reviewer 2 Report
Here the authors investigate a possible role of lysosomes in myogenic cells and a possible interaction with canonical Wnt signalling. They characterise lysosomes in primary cells isolated from embryonic chicken breast muscle using dyes and antibody staining. The cultured cells include mono nucleated myoblasts, myotubes and fibroblasts and parameters such as lysosome size and sub cellular localisation are examined, as well as effects of pharmacological activation of canonical Wnt signalling.
While this is an interesting topic that is would benefit from further study, in my view the data are not well presented and the conclusions are not convincingly supported by what is shown.
Fig. 1 the image seems blurry, also add scale bars for magnified panels, measurement of "area" does not seem appropriate, this should be volume. Make clear how this was assessed 'per cell', was there a membrane marker used? (also relevant to Fig. 4)
Fig. 2 add labelling on panels, are these different time points, different magnifications - this is not clear, also the legend states that LAMP2 labeling was found within the nuclei - this is difficult to see and not convincing, please provide 3D reconstruction z-stack to confirm this conclusion
Fig. 3 add scale bar to magnified panels, visually there appears to be a difference, but this needs to be quantified
Fig. 4 the statements in the text are not supported by what is shown, for example it is difficult to see that BIO/Wnt3a reverts effects of Lys05
Fig. 6 no scale bars, no quantification, nuclear localisation of b-catenin needs to be shown more clearly, what do the arrows indicate?
Author Response
Dear Ms. Kesic,
We submitted the manuscript entitled "The Wnt/beta-catenin pathway and cytoskeletal filaments are involved in the positioning, size, and function of lysosomes during chick myogenesis" (cells-1962844) by Bagri et al., for publication in Cells. The reviewers recommended major revisions to our manuscript. We made the modifications in the manuscript and in its figures and we included a point-by-point response to the reviewer’s comments, which were useful to improve the manuscript. We thank the reviewers for their careful and appropriate analysis. The detailed corrections are as follows.
Reviewer: 1
1 - The topic is highly relevant to the scientific field and to the technological advance for the development of new therapies for human and animal health. The article is well-written, clear, and the experimental design is adequate. Therefore, I recommend accepting the manuscript for publication.
Author’s response: We thank the reviewer for the analysis and comments.
Reviewer: 2
1 - Here the authors investigate a possible role of lysosomes in myogenic cells and a possible interaction with canonical Wnt signaling. They characterize lysosomes in primary cells isolated from embryonic chicken breast muscle using dyes and antibody staining. The cultured cells include mono nucleated myoblasts, myotubes and fibroblasts and parameters such as lysosome size and sub cellular localization are examined, as well as effects of pharmacological activation of canonical Wnt signaling. While this is an interesting topic that is would benefit from further study, in my view the data are not well presented, and the conclusions are not convincingly supported by what is shown.
Fig. 1 the image seems blurry, also add scale bars for magnified panels, measurement of "area" does not seem appropriate, this should be volume. Make clear how this was assessed 'per cell', was there a membrane marker used? (Also relevant to Fig. 4).
Author’s response: We thank the reviewer for the comments. We now changed Figure 1 by adding scale bars in all images and by improving the quality and resolution of the images. We measured the total area occupied by lysosomes in the three phenotypes of cultured cells (myoblasts, myotubes and fibroblasts) by analyzing images of Lysotracker (lysosomal marker), Desmin (muscle marker), Hoechst (nuclear marker) and phase contrast microscopy (whole cell image). We did not use confocal microscopy to analyze the 3D volume of lysosomes. The limits of cells were accessed by phase contrast microscopy. We now corrected and improved the description on how we measured lysosomes in the new version of the manuscript.
2 - Fig. 2 add labelling on panels, are these different time points, different magnifications - this is not clear, also the legend states that LAMP2 labeling was found within the nuclei - this is difficult to see and not convincing, please provide 3D reconstruction z-stack to confirm this conclusion.
Author’s response: We now changed Figure 2 by adding scale bars and labeling in all images. All three images are representatives of chick myogenic cells grown in culture for 48 hours. We removed the sentence related to the nuclear localization of LAMP2, since we did not confirm these results in 3D analysis of confocal z-stacks.
3 - Fig. 3 add scale bar to magnified panels, visually there appears to be a difference, but this needs to be quantified.
Author’s response: We now changed Figure 3 by adding scale bars in all images.
4 - Fig. 4 the statements in the text are not supported by what is shown, for example it is difficult to see that BIO/Wnt3a reverts effects of Lys05.
Author’s response: We agree with the reviewer that it was difficult to see the results by showing only the quantifications of the effects of Lys05, BIO and Wnt3a. Therefore, we now included representative images of all the experimental conditions showing the effects of all cell treatments by immunofluorescence microscopy.
5 - Fig. 6 no scale bars, no quantification, nuclear localization of b-catenin needs to be shown more clearly, what do the arrows indicate?
Author’s response: We now changed Figure 6 by adding scale bars in all images, an explanation about what the arrows indicate and quantifications of the nuclear localization of beta-catenin and myotube thickness.
Reviewer: 3
1 - In this manuscript, the authors investigated the morphological and functional characteristics of lysosomes in muscle and non-muscle cells during chick myogenesis. They found that the positioning of lysosomes is distinct in different cell types: in fibroblasts, the lysosomes are closer to the plasma membrane while are near the nucleus in myoblasts and myotubes. Furthermore, they showed that the distribution and size of lysosomes could be regulated by manipulating either Lamp2 or Wnt/beta-catenin pathway. More importantly, the effects on myogenesis by knocking-down LAMP2 can be rescued by the activation of the Wnt/beta-catenin pathway, suggesting a strong connection between these two pathways. This study provides some insight into the function of lysosomes in skeletal muscle differentiation and explores the potential pathway involved. Overall, the conclusion is largely supported by the data. However, there are still several questions that need to be addressed before it can be accepted for publication.
Fig. 1 and 2, the quality of the images is too low to evaluate. Basically, it appeared that one cell only has a red spot lysosome based on the images provided in this version. A higher resolution of images needs to be shown to make the conclusion solid.
Author’s response: We thank the reviewer for the comments. We now improved the quality and resolution of the images shown in Figures 1 and 2.
2 - Fig. 6, the authors need to include a western blot to show the knockdown efficacy of the Lamp2 by the siRNA. Also, to rule out the potential off-target effect, it will be beneficial to show that overexpression of Lamp2 can rescue the phenotype caused by siRNA knockdown.
Author’s response: A western blot showing the efficacy of the knockdown of LAMP2 by the same siRNA sequence was shown in a previous paper from our lab (Figure 3 in Bagri et al., 2020, doi.org/10.1155/2020/6404230). In this previous work we showed that transfection of chick muscle cells with siRNA against LAMP2 induced an 72% reduction in LAMP2 protein levels. We now included the following text in the new version of our manuscript: “Chick myogenic cells were grown for 24 hours and transfected with siRNA against LAMP2. Our group have previously shown the efficacy of the knockdown of siRNA sequences specific for LAMP2 of Gallus gallus in a previous paper from our lab [13]. In this previous work we showed by Western Blot that transfection of chick muscle cells with siRNA against LAMP2 induced ~70% reduction in LAMP2 protein levels”. We appreciated the suggestion to include data showing that overexpression of LAMP2 can rescue the phenotype caused by siRNA knockdown. Unfortunately, these experiments will demand an extension in the deadline for this review since our country is experiencing major problems in the acquisition of reagents for research (mainly due to the COVID pandemic crisis). If the reviewer consider that these experiments are essential, we could try to perform them.
3 - Fig. 6, to make an argument that activation of the Wnt/beta-catenin pathway can rescue Lamp2 knockdown, the authors should include quantification.
Author’s response: We now included a quantification of the levels of beta-catenin in the cytoplasm and nucleus and myotube thickness.
We hope that the modifications made in the new version of the manuscript have properly addressed the criticism and suggestions made by the referees, and that the improvements made in the manuscript will be sufficient for its publication in Cells.
We would like to state that all listed authors qualify for authorship and agreed in the submission of the manuscript. The final version of the manuscript has been seen and approved by all coauthors. The authors declare that they have no conflict of interest.
With kind regards,
Prof Claudia Mermelstein

Reviewer 3 Report
In this manuscript, the authors investigated the morphological and functional characteristics of lysosomes in muscle and non-muscle cells during chick myogenesis. They found that the positioning of lysosomes is distinct in different cell types: in fibroblasts, the lysosomes are more close to the plasma membrane while are near the nucleus in myoblasts and myotubes. Furthermore, they showed that the distribution and size of lysosomes could be regulated by manipulating either Lamp2 or Wnt/beta-catenin pathway. More importantly, the effects on myogenesis by knocking-down LAMP2 can be rescued by the activation of the Wnt/beta-catenin pathway, suggesting a strong connection between these two pathways. This study provides some insight into the function of lysosomes in skeletal muscle differentiation and explores the potential pathway involved. Overall, the conclusion is largely supported by the data. However, there are still several questions that need to be addressed before it can be accepted for publication.
1. Fig.1 and 2, the quality of the images is too low to evaluate. Basically, it appeared that one cell only has a red spot lysosome based on the images provided in this version.
A higher resolution of images need to be shown to make the conclusion solid.
2. Fig.6, the authors need to include a western blot to show the knockdown efficacy of the Lamp2 by the siRNA. Also, to rule out the potential off-target effect, it will be beneficial to show that overexpression of Lamp2 can rescue the phenotype caused by siRNA knockdown.
3. Fig.6, to make an argument that activation of the Wnt/beta-catenin pathway can rescue Lamp2 knockdown, the authors should include quantification.
Author Response

(The authors gave the same response as above.)

Round 2
Reviewer 2 Report
The authors provide better images and additional quantifications. This has improved the manuscript.
Please check labelling of panels, e.g. fig 4 I cannot see Wnt3a plus Lys05, or BIO plus Lys05
Author Response
Dear Ms. Kesic,
We submitted a revised version of the manuscript entitled "The Wnt/beta-catenin pathway and cytoskeletal filaments are involved in the positioning, size, and function of lysosomes during chick myogenesis" (cells-1962844) by Bagri et al., for publication in Cells. One reviewer recommended minor revisions to our manuscript. We made the modifications in the manuscript and in one figure and we included a point-by-point response to the reviewer’s comments. We thank the reviewer the careful and appropriate analysis. The detailed corrections are as follows.
Reviewer: 2
1 - The authors provide better images and additional quantifications. This has improved the manuscript. Please check labelling of panels, e.g. fig 4 I cannot see Wnt3a plus Lys05, or BIO plus Lys05.
Author’s response: We thank the reviewer for this important comment. We now corrected the labelling of all panels from Figure 4.
We hope that the modifications made in the new version of the manuscript have properly addressed the comments made by the reviewer, and that the improvements made in the manuscript will be sufficient for its publication in Cells.
With kind regards,
Prof Claudia Mermelstein

Reviewer 3 Report
In this revision, the authors addressed most of the concerns, now the manuscript is suitable for publication.
Author Response
Author’s response: We thank the reviewer for the analysis and comments.
With kind regards,
Prof Claudia Mermelstein